# Mechanical, Thermal, and Moisture Buffering Properties of Novel Insulating Hemp-Lime Composite Building Materials

**DOI:** 10.3390/ma13215000

**Published:** 2020-11-06

**Authors:** Yaser Abdellatef, Mohammad Amil Khan, Asif Khan, Mehdi Iftekharul Alam, Miroslava Kavgic

**Affiliations:** 1Civil Engineering Department, University of Ottawa, 161 Louis Pasteur, Ottawa, ON K1N 6N5, Canada; yabde079@uottawa.ca; 2Mechanical Power Department, Faculty of Engineering, Cairo University, Giza 12613, Egypt; 3Civil Engineering Department, University of Manitoba, 15 Gillson St., Winnipeg, MB R3T 5V6, Canada; mohammadamilkhan@gmail.com (M.A.K.); asifkhantupu@gmail.com (A.K.); mehdi17uofm@gmail.com (M.I.A.)

**Keywords:** hemp-lime composites, mechanical properties, thermal properties, moisture buffering capacity, crushed brick, metakaolin, hydrated lime, hydraulic lime

## Abstract

Hempcrete is a sustainable biocomposite that can reduce buildings’ embodied energy while improving energy performance and indoor environmental quality. This research aims to develop novel insulating hemp-lime composites using innovative binder mixes made of recycled and low-embodied energy pozzolans. The characterization of composites’ mechanical and hygrothermal properties includes measuring compressive strength, splitting tensile strength, thermal conductivity, specific heat capacity, and moisture buffer capacities. This study also investigates the impact of sample densities and water content on compressive strength at different ages. The findings suggest that mixes with a 1:1 binder to hemp ratio and 300−400 kg/m^3^ density have hygrothermal and mechanical properties suitable for insulating infill wall applications. Hence, compressive strengths, thermal conductivity, and specific heat capacity values range from 0.09 to 0.57 MPa, 0.087 to 0.10 W/m K, and 1250 to 1557 J/kg K, respectively. The average moisture buffer value for all hempcrete samples of 2.78 (gm/m^2^ RH%) indicates excellent moisture buffering capacity. Recycled crushed brick pozzolan can enhance the hygrothermal performance of the hemp-lime composites. Thus, samples with 10% crushed brick have the lowest thermal conductivity considering their density and the highest moisture buffer capacity. The new formulas of hydrated lime and crushed brick have mechanical properties comparable to metakaolin and hydraulic lime formulas.

## 1. Introduction

Hempcrete, made of the hemp plant’s inner woody core mixed with a lime-based binder, is a promising lightweight biocomposite material that holds the potential to considerably improve energy and indoor environmental performance and sustainability of buildings [1,2]. Nevertheless, the utilization of hempcrete in the construction industry remains low, mainly due to the high variability of hemp-lime composites and lack of formulas that use widely available and locally sourced ingredients. For example, hempcrete’s production cost and embodied energy could be considerably higher in Canada than in Europe due to imported materials [3]. The use of innovative binders and a better understanding of the material’s performance are indispensable steps for addressing hempcrete underutilization.

Previous studies have shown that hempcrete hygrothermal and mechanical properties depended on various factors. The binder to hemp ratio, compaction level, water content, and binder mix design have been considered to be critical [4,5,6,7]. For instance, an increase in the proportion of porous hemp hurd in hempcrete mixture reduced the dry density and the thermal conductivity [6,8]. In contrast, there was a positive correlation between density and thermal conductivity of the hempcrete [9,10]. Thus, studies have reported thermal conductivity in the range of 0.06−0.11 W/m K for hempcrete mixtures with dry densities ranging from 200 to 400 kg/m^3^ and hemp/binder ratios of 1:1, 1:1.5, and 1:2 [11,12]. In comparison, hempcrete samples with higher dry density and hemp/binder ratio in the range of 450−800 kg/m^3^ and 1:2−1:4, respectively, had higher thermal conductivity ranging from 0.12 to 0.18 W/m K [11,12]. Previous findings have also indicated a negative correlation between hempcrete’s density and moisture buffering capacity. For example, Latif et al. (2015) reported a moisture buffer value (MBV) of 3.47 gm/m^2^%RH with a binder to hemp ratio of 1.2:1, and density of 290 kg/m^3^ [13], while Collet et al. (2013) reported an MBV of 2.14 gm/m^2^%RH, with a binder to hemp ratio of 2:1, and density of 430 kg/m^3^ [14].

Furthermore, an increase in water content tends to increase the thermal conductivity of the hempcrete mixes. Rahim et al. (2016) reported a rise in thermal conductivity of approximately 8% and 16% at 10 °C and 40 °C, respectively, due to an increase in the water content of 0.08 kg/kg from the dry state [15]. The variation in specific heat capacity and thermal diffusivity under different humidity conditions was considerably less apparent than the variation in thermal conductivity [8]. Thus, specific heat capacity in the range of 1000−1590 J/kg K was reported for hempcrete with dry densities ranging from 381 to 627 kg/m^3^ and hemp to binder ratio 1:2 [5,10]. Thermal diffusivity in the range of 1.48 × 10^−7^ m^2^/s in the dry state to 0.98 × 10^−7^ m^2^/s in the fully saturated condition and high specific heat capacity can provide a better thermal performance than suggested by hempcrete’s thermal transmittance [16]. Moreover, hempcrete integration with other innovative materials can modify its thermal energy storage properties [17].

The typical value of the compressive strength of hempcrete mixes is between 0.2 and 1.15 MPa, whereas their flexural and splitting tensile strength values are in the range of 0.06−1.3 MPa and 0.02−0.15 MPa, respectively [9,18]. A high content of hemp hurd in the design mix reduces the compressive and flexural strength of hemp-lime composites [9,19]. However, an increase in the density through compaction of the hempcrete can enhance its mechanical characteristics while reducing the binder’s quantity, which is the major contributor to embodied carbon [18,20,21]. In this respect, Dinh (2014) reported approximately 15% higher compressive strength of a 556.0 kg/m^3^ hempcrete sample than a 439.7 kg/m^3^ sample with a compressive strength of 0.65 ± 0.06 MPa [6]. Likewise, the hempcrete mixes’ splitting tensile strength increased with their density [18]. For instance, Elfordy et al. (2008) reported flexural strength in the range of 0.749−1.209 MPa, with a 2:1 binder to hemp ratio and a high-density range of 430−607 kg/m^3^ [18].

Calcium hydroxide (i.e., hydrated lime) is particularly suitable for hemp-lime composites’ production due to advantages such as breathability, high water vapour permeability, durability, mould resistance, and pest deterrence [22,23]. Furthermore, adding hydraulic lime and metakaolin can improve hempcrete’s performance due to their fast setting, high reactivity, and lower embodied energy as compared with Portland cement [22,24]. Crushed brick is another promising pozzolan that belongs to an aluminosilicate group and is reactive towards lime, creating interfacial surface alkaline [25]. Therefore, utilizing reclaimed bricks from demolished buildings as pozzolanic lime mortars could extend their lifetime and reduce hempcrete’s embodied energy.

This research study aims to develop innovative insulating hemp-lime composites made of recycled and locally sourced materials. To the best of the authors’ knowledge, this is the first study that has used recycled crushed brick as a pozzolan to increase hemp-lime composites’ availability and performance, while reducing their environmental impact. Unlike most previous studies that focused on hempcrete’s mechanical, thermal, or moisture properties [10,23], this study experimentally estimated the mechanical strength (compressive and splitting tensile strength), thermal performance, and moisture regulation capabilities of eight hempcrete design mixes. Furthermore, in contrast to the existing studies that utilized a hemp/binder ratio of 1:2 or higher for wall formulas [18,26], this research focused on maximizing the hemp hurd ratio to improve the thermal properties of the hempcrete mixtures. Therefore, a hemp/binder ratio of 1:1 with a targeted density of 300 to 400 kg/m^3^ was applied to create an insulating ”infill wall” formula. The effect of water content on the mechanical properties was also studied. Consequently, this research provides valuable information about insulating hemp-lime composites’ critical performance required for material characterization that is likely to interest audiences in academia and industry, focusing on sustainable low-carbon composite building materials.

## 2. Materials and Methods

### 2.1. Raw Materials

Local hemp producer “Plains Hemp” provided the hemp hurd used as aggregate in this study. The bulk density of the hurd at room temperature of 20 °C was 110 ± 5 kg/m^3^, whereas its dry density after oven drying at 105 °C for 24 hours was 101 ± 2 kg/m^3^. Sieve analysis was performed on an oven-dried batch of 100 g, according to ASTM C136–06 [27], to determine the distribution of fine and coarse aggregates. Approximately 87% of the particles were less than 2.36 mm, and 10% were between 2.36 and 6.3 mm. The hydrated lime, metakaolin, and hydraulic lime, used in this study complied with ASTM C207−06 [28], ASTM C618−12 [29], and ASTM 1707−09 [30], respectively. Utilized recycled pozzolanic crushed brick had a density of ~1355 kg/m^3^, firing temperature between 600 and 900 °C, and it was primarily composed of silicon dioxide (50−60%) and aluminum oxide (20−30%).

### 2.2. Compositions of Hempcrete Samples

This study focused on maximizing the hemp hurd ratio within the hempcrete mixture to improve its thermal properties while reducing its environmental impact. Hence, the hemp hurd to binder ratio used in the hempcrete sample preparation was 1:1 with a wall application density of 300 and 400 kg/m^3^ [5,31]. Table 1 summarizes eight hempcrete design mixes with varying pozzolan concentrations and fifteen batches with varying water contents. The amount of water added to hemp-lime composites affects their physical properties. For instance, low water content may cause incomplete hydration and unreacted binder in powder form, leading to reduced compressive strength [23,32]. In contrast, high water content can result in higher density, leading to an increase in thermal conductivity, issues with a setting for the lime binder, and excessively long drying times for the hempcrete [6,33]. Furthermore, there is a lack of tests for measuring the workability of hemp-lime concretes, and experience is used to determine the appropriate water content. Therefore, we investigated different water contents that allowed the suitable workability of the developed hempcrete mixes.

### 2.3. Mixing, Moulding, and Curing

Mixing binders and water to create a slurry was the first step in creating hempcrete samples. Then, hemp hurd was added to the slurry and blended in an industrial mixer for approximately 5 min until a uniform mixture was obtained. For the thermal properties’ tests and moisture buffering tests, rectangular wooden moulds (26 cm wide × 26 cm long × 5.5 cm high) were filled with the mixture and tamped until reaching the desired wet density. The cylindrical moulds (10 cm diameter × 20 cm long) for compression tests were prepared according to ASTM C39/C39M−12 [34]. Furthermore, for splitting tensile tests, the cylindrical moulds (15 cm diameter × 30 cm long) were produced by following ASTM C496/C496M–17 [35]. Then, a one-quarter portion of the mould at a time was filled with the mixture and tamped until the desired wet density was reached (i.e., 750−800 kg/m^3^). The samples were demolded after seven days and cured at room temperature of 22 ± 1 °C, with RH ~50% for another 21−28 days until they reached a constant mass. In total, approximately 220 samples were tested for their mechanical, thermal, and moisture buffering properties.

### 2.4. Experimental Tests

#### 2.4.1. Mechanical Tests

The compression and splitting tensile tests were performed by following the ASTM D4832 [36] and ASTM C496 [35] standards, respectively. The compressive strength was measured after 28, 60, and 90 days, while splitting tensile strength was measured after 90 days. The load was applied continuously without any shock at a 5 mm/min rate and run for approximately 4−5 min per specimen. The mode of failure was determined by the drop observed in the load-displacement curve after the initial increase. For each design mix and age, we tested four samples and calculated compressive strength as their average. Data obtained from the compression test were used to calculate Young’s modulus, according to ASTM C469 [37]. For splitting tensile tests, we tested two samples for each design mix and computed their averages.

#### 2.4.2. Thermal Tests

A FOX 314 Heat Flow Meter Apparatus (HFMA) was used for thermal conductivity and specific heat capacity measurements of the hempcrete samples, according to ASTM C518 [38] and ASTM C1784 [39] standards, respectively. Thermal tests included ”wet” samples at room temperature of 22 ± 1 °C, with RH ~50% and ”dry” samples, oven-dried at 100 ± 5 °C until a constant mass was reached (i.e., changes in weight readings of less than 0.1%). The HFMA created steady one-dimensional heat flux through the specimens by setting both plates at constant but different temperatures with an accuracy of ± 0.03 °C. The HFMA applied Fourier’s law of heat conduction to obtain hempcrete’s thermal conductivity with an accuracy of approximately 1% at a specific mean temperature of the plates. Additionally, we calibrated the HFMA to convert transducers’ voltage signals to heat fluxes using NIST 1450b SRM (Standard Reference Material of the National Institute of Standards and Technology) [40]. The HFMA was modified to measure hempcrete samples’ specific heat capacity by first maintaining both plates at the equal temperature until reaching a steady-state condition, then changing the plates’ temperatures by the same step, and holding them until reaching a new steady-state condition. By assuming transient heat conduction in a finite body for a significant time, the specific heat capacity was determined at each temperature step using Equation (1), according to Tleoubaev and Brzezinski (2007) [41]:(1)$$C_{p}=\frac{k}{\rho L^{2}\;\pi^{-2}\left(\frac{\Delta ln\left(Q_{i}\right)}{\Delta t}\right)}$$
where *L* is sample thickness (m), *ρ* is density (kg/m^3^), *k* is thermal conductivity (W/m K), and (Δ*lnQ_i_*_/_Δ*t*) is the slope of the curve of the natural logarithm of HFMA output values versus time.

The uncertainty in specific heat capacity was calculated at each temperature setpoint to be approximately ± 2.5%, using Equation (2):(2)$$U_{C_{p}}=\sqrt{{\left(\frac{\partial C_{p}}{\partial k}U_{k}\right)}^{2}+{\left(\frac{\partial C_{p}}{\partial \rho }U_{\rho }\right)}^{2}+{\left(\frac{\partial C_{p}}{\partial L}U_{L}\right)}^{2}+{\left(\frac{\partial C_{p}}{\partial \left(slope\right)}U_{Slope}\right)}^{2}}$$
where *U_k_* is uncertainty in thermal conductivity ( ± 1%), *U_ρ_* is uncertainty in density (±1.75%), *U_L_* is uncertainty in thickness (±0.025 mm), and *U_slope_* is uncertainty in slope (±0.4% with 95% confidence level).

Furthermore, the hempcrete samples’ thermal diffusivity was calculated using HFMA measurements of thermal conductivity and specific heat capacity, as shown in Equation (3):(3)$$\alpha =\frac{k}{\rho \;C_{p}}\;$$
where *k* is the thermal conductivity (W/m K), ρ is density (kg/m^3^), and *C_p_* is the specific heat capacity (J/kg K).

#### 2.4.3. Moisture Buffering Capacity Tests

According to the NORDTEST protocol [42], the measurements of hempcrete’s moisture buffer capacity included sealing all specimens’ sides except the upper surface (26 cm × 26 cm), exposed to humidity cycles. The air temperature in the climate chamber was kept constant at 23 ± 1 °C, while the relative humidity (RH) was kept constant at 75 ± 2% for 8 hours and 33 ± 2% for 16 hours. Due to the importance of the initial conditions of the samples’ moisture contents [43,44], their effect was studied using two different initial states, high moisture content (75% RH) and low moisture content (33% RH). The mass of each sample mass was measured using an electronic scale with an accuracy ± 0.01 gm to monitor the mass change during the uptake and release phases. The humidity cycles were repeated until the steady-state condition was reached, and the mass change in the cycles’ uptake and release phases was less than 5% [42]. The moisture buffer value (MBV) was calculated as the average of the last three steady cycles, using Equation (4):(4)$$\mathrm{MBV}\;=\frac{\Delta m}{A\;(\Delta \mathrm{RH}\;\%)}$$
where Δ*m* in the mass change (gm), A is the exposed surface area (m^2^), and Δ*RH*% is the difference between the high and low relative humidity.

## 3. Results

### 3.1. Density Results

Figure 1 shows that approximately 91% of the specimens are within the density range from 300 to 350 kg/m^3^. The average density and the amount of variation are 323 kg/m^3^ and 12 kg/m^3^, respectively, the minimum is 290 kg/m^3^, and the maximum is 372 kg/m^3^. These results suggest consistency between the average densities of hempcrete samples produced in this study. Furthermore, Table 2 summarizes descriptive statistics of compression, tensile, and thermal samples. The samples prepared for compressive tests have the highest average density (⁓340 kg/m^3^), followed by the splitting tensile samples (⁓324 kg/m^3^), and thermal samples (⁓315 kg/m^3^). Similarly, the specimens created for the compressive tests exhibit the highest standard deviation of 17.2 kg/m^3^ (⁓5%), followed by tensile specimens with a standard deviation of 12.1 kg/m^3^ (⁓3.8%). Thermal samples share similar variations in density (⁓2.7%), for both wet and dry tests.

### 3.2. Compression and Splitting Tensile Test Results

Table 3 summarizes the average compressive strength, splitting tensile strength, modulus of elasticity, and hempcrete samples’ densities. The samples’ average compressive strength is 0.28 MPa, with a standard deviation of 0.098 MPa (35%), and it ranges from 0.11 MPa (LMK70B) to 0.47 MPa (LMK50B). Due to less variation in the samples’ density for splitting tensile tests than compressive tests (see Table 2), tensile strength dispersion is slightly lower than compression values. Thus, the average splitting tensile strength of all samples is 0.024 MPa, with a standard deviation of 0.0065 MPa (27%), and it ranges from 0.0101 MPa to 0.0348 MPa.

Overall, an increase in hempcrete density leads to a rise in its compressive and tensile strengths. These findings are consistent with the results of previous studies [18]. Thus, the hempcrete samples of 350 kg/m^3^ and denser have an average compressive strength of 0.33 MPa and higher. In contrast, the specimens with the lowest density, including LMK70B (303.5 kg/m^3^), LMK70 (318.8 kg/m^3^), and LMK70C (327.1 kg/m^3^), have the lowest average compressive strength ranging from 0.11 MPa to 0.18 MPa. Similarly, the highest splitting tensile strength of 0.0348 MPa has the LNHL50B sample with the highest density of 339.15 kg/m^3^, whereas the least dense LMK70B (⁓289 kg/m^3^) has the lowest splitting tensile strength of 0.01 MPa. Nevertheless, the two samples with the highest compressive strengths, LMK50B, and LNHL50B, are not the most compacted hempcrete samples. These results suggest that water content, binder type, and content also significantly impact hemp-lime concretes’ mechanical properties.

Hence, on the one hand, an increase in the water content increases the compressive strength, even of the samples with lower densities. For example, the LCB20 sample with 2.5 kg of water and 344.1 kg/m^3^ density has a compressive strength of 0.19 MPa, while the LCB20B sample with 3 kg of water and 323.8 kg/m^3^ density has a compressive strength of 0.26 MPa. Metakaolin samples with percentages above 20%, and all LNHL samples exhibit a similar trend. On the other hand, the design mixes with smaller water content (i.e., 2.5:1 water to hemp ratio) show inferior compressive and tensile strengths compared with other hempcrete specimens. These results echo Walker and Pavia’s (2014) observations regarding hemp’s high suction ability that undermined the binder’s hydration and adversely affected the strength of hemp-lime composites [23]. The sample produced from LMK70 (lime 30% and metakaolin 70%) show the highest amount of unreacted binder and has the lowest compressive and splitting tensile strengths. The LNHL50 sample, cast in two batches, shows an average compressive strength of only 0.26 MPa using a water to hemp ratio of 2.5:1 and approximately 60% higher (0.42 MPa) using a water to hemp ratio of 3:1. Similarly, the LNHL70 sample prepared with a water to hemp ratio of 2.5:1 exhibits an average compressive strength of 0.30 MPa, which is around 21% below the compressive strength of the sample with water to hemp ratio of 3:1.

Furthermore, an increase in the percent of metakaolin and crushed brick above a certain amount reduces the compressive strength of hempcrete. This effect is most pronounced in the LMK samples in which compressive strength decreases from 62% to 76%, with an increase in the percent share of metakaolin from 50% to 70%. Similarly, an increase in the percent share of metakaolin and NHL hurts the tensile strength of hempcrete. Thus, an increase in the percentage of metakaolin from 50% to 70% results in a 16% to 62% decrease in hempcrete tensile strength, depending on the batch. An increase in hydraulic lime from 50% to 70% decreases the tensile strength for approximately 15% to 34%, whereas an increase in the crushed brick from 10% to 20% exhibits the lowest decreases in the tensile strength of 8% to 10%. On the one hand, these results support Dinh (2014) findings regarding the adverse effect of an increase in metakaolin share above 70% on the compressive strength of hempcrete [6]. On the other hand, Eires et al. (2006) stated that 75% of metakaolin and 25% of lime were the best binder mix [45]. However, they also reported that this design mix required higher curing temperatures to increase strength gain.

As presented in Table 3, the modulus of elasticity shows a wide variation between design mixes and a weak relationship with the samples’ density. Thus, Young’s modulus ranges from 3.16 to 10.84 MPa, and its average value of all samples is 7.46 MPa, with a standard deviation of 2.327 MPa (31%). Increasing the metakaolin and crushed brick content reduces Young’s modulus, whereas increasing hydraulic lime does not have the same effect. The LCB20 sample shows the lowest value of Young’s modulus (3.16 MPa), followed by LMK70C (4.00 MPa). In contrast, the LMK10 specimen has the highest Young’s modulus (10.84 MPa), followed by LCB10 (10.42 MPa) and LNHL70 (10.22 MPa).

Figure 2 compares densities and compressive strengths on the 28th, 60th, and 90th day among the developed samples and against previous studies with similar hempcrete densities [18,26]. The results show that specimens, on the one hand, with the lowest densities of approximately 303 kg/m^3^ and 318 kg/m^3^ have 30–50% lower compressive strength than the results in the literature [18,26]. The likely reason is a higher content of binder used in these studies. In this regard, Elfordy et al. (2008) [18] and Williams et al. (2017) [26] used a 2:1 and 2.25:1, respectively, binder to hemp ratio. On the other hand, samples with a density of approximately 340 kg/m^3^ and above have compressive strength comparable with Elfordy et al. (2008) [18] and Williams et al. (2017) [26]. Some studies have also differentiated between testing in the perpendicular (⊥) and the parallel (∥) direction to the direction of compaction. In this respect, Williams et al. (2017) reported slightly higher compressive strengths of the samples with similar density tested in perpendicular than parallel direction (see Figure 2) [26]. In this research, the compaction and testing directions (i.e., perpendicular or parallel) were designed to present the worst-case scenario for mechanical properties. Therefore, parallel testing to the compaction direction might be another reason for differences between our results and values reported by Williams et al. (2017) [26].

### 3.3. Thermal Properties Results

Table 4 presents the dry and wet hempcrete samples’ thermal properties, including their density, thermal conductivity, specific heat capacity, and thermal diffusivity. The wet hempcrete samples have higher density, thermal conductivity, and specific heat capacity than their dry counterparts due to the higher moisture content. Thus, the wet density ranges from 304 kg/m^3^ to 336 kg/m^3^, and the average is 322 kg/m^3^. In comparison, the dry density is approximately 4.5% lower, ranging from 291 kg/m^3^ to 321 kg/m^3^ with an average of 307 kg/m^3^. The thermal conductivities of wet and dry samples vary from 0.091 to 0.101 W/m K and from 0.087 to 0.096 W/m K, with averages of 0.096 and 0.0913 W/m K, respectively. Furthermore, the specific heat capacities of wet and dry samples range from 1398 to 1557 J/kg K and from 1250 to 1421 J/kg K, with averages of 1508 and 1365 J/kg K, respectively. These results range from specific heat capacity values reported in the literature, between 1000 and 1560 J/kg K [5,46].

In addition, the wet samples’ thermal diffusivity is lower as compared with their dry counterparts. In this respect, thermal diffusivity of the wet samples ranges from 1.893 to 2.069 (m^2^/s) × 10^−7^, whereas dry specimens range from 2.076 to 2.321 (m^2^/s) × 10^−7^, with averages of 1.970 (m^2^/s) × 10^−7^ and 2.182 (m^2^/s) × 10^−7^, respectively. These findings indicate that, in the wet state, the total increase in heat storage $\left(\rho \times \;c_{p}\right)$ is higher than the increase in thermal conductivity. Thus, the wet state’s thermal diffusivity $\left(\frac{k}{\rho \;C_{p}}\;\right)$ is lower than that of the dry state (i.e., have higher thermal inertia than the dry samples.) Our results are slightly higher than the range of thermal diffusivity values reported in the literature, between 0.98 and 1.68 (m^2^/s) × 10^−7^ [5,47]. A possible explanation might be the higher density values that previous studies used to calculate thermal diffusivity.

The results presented in Table 4 indicate that the hempcrete samples with crushed brick have the lowest thermal conductivity and specific heat capacity, considering their density and the highest thermal diffusivity. For example, the average density, thermal conductivity, specific heat capacity, and thermal diffusivity of all hempcrete samples with crushed brick (wet and dry) are approximately 318 kg/m^3^, 0.93 W/m K, 1376 J/kg K, and 2.142 (m^2^/s) × 10^−7^, respectively. In comparison, the metakaolin hempcrete samples have a similar average density (⁓317 kg/m^3^) but higher thermal conductivity (⁓0.95 W/m K) and specific heat capacity (⁓1450 J/kg K). The LNHL hempcrete samples have the lowest average density of approximately 307 kg/m^3^, and therefore the lowest thermal conductivity of 0.091 W/m K. Nevertheless, they have the highest average specific heat capacity of 1470 J/kg K, and the lowest thermal diffusivity of 2.020 (m^2^/s) × 10^−7^.

Figure 3 illustrates the relation between thermal conductivity and density in the dry and wet states for all design mixes. Similar to previous studies [9,26], both the dry and wet samples show a positive linear relationship between the samples’ conductivities and densities with R^2^ values of 0.907. Thus, the LNHL50 samples with the lowest dry and wet densities of 291 and 304 kg/m^3^ have the lowest thermal conductivity of approximately 0.087 and 0.90 W/m K, respectively. In contrast, the LMK10 samples with the highest dry and wet densities of 321 and 336 kg/m^3^ have the highest thermal conductivity of approximately 0.096 and 0.1 W/m K, respectively.

Figure 4 compares the experimental average thermal conductivities of hempcrete samples against the results of previous studies. The results show that the thermal conductivities and their increase with density are comparable to the values reported by previous studies for the specimens with similar densities. Furthermore, some studies differentiate between testing in the perpendicular (⊥) and parallel (∥) direction (i.e., heat flux direction) to the compaction of the samples. In this respect, Williams et al. (2017) [26] and Nguyen et al. (2010) [48] reported higher thermal conductivity of the samples with similar density tested in perpendicular than parallel direction (see Figure 4). Moreover, Nguyen et al. (2010) [48] reported a higher difference between the parallel and perpendicular conductivities than Williams et al. (2017) [26]. The likely reason is the higher density range of the samples in the first study compared with that of the second study. In this research, the compaction and testing directions (i.e., perpendicular or parallel) represent the worst-case scenario for the thermal properties. In this respect, the HFMA created heat flux across the samples, perpendicular to the compaction direction. Thus, our results show a good match with experiments conducted in the perpendicular direction [26], and a more significant discrepancy with those performed in the parallel orientation [26].

### 3.4. Moisture Buffering Results

Figure 5 presents a comparison of moisture buffer values (MBV) for each cycle’s uptake and release phases until reaching the steady-state of three selected hempcrete samples, LMK50, LNHL50, and LCB10, with two initial moisture conditions, high and low. As shown for the high moisture initial condition, the MBV for the release phase starts with a higher value (2.4−2.6 gm/m^2^ RH%) than that of the uptake phase (1.3−1.7 gm/m^2^ RH%). The hempcrete samples tend to lose the high moisture content obtained from the initial condition more than absorbing more moisture. However, after the initial moisture content reduces, the MBV for the uptake phase increases again. In contrast, the MBV for the release phase slightly decreases until both reach steady-state starting from the fourth cycle with less than 5% difference between MBV for uptake and release. For the low moisture initial condition case, the MBV for the uptake phase starts with a higher value (3−4 gm/m^2^ RH%) than that of the release phase (2−2.8 gm/m^2^ RH%). The hempcrete samples are capable of absorbing more moisture due to the low initial moisture content. The MBV for the uptake phase decreases again, and the MBV for the release phase slightly increases until both reach steady-state starting from the fourth cycle, with a difference of less than 5% between MBV for uptake and release.

The three design mixes express different moisture buffering behaviors. For example, the LNHL50 with high initial moisture condition starts with 14% and 26% higher MBV as compared with LCB10 and LMK50, in the uptake phase and 2.6% and 13.8% higher in the release phase, respectively. Furthermore, the starting MBV values of the LCB10 with low initial conditions are 11.2% and 39% higher than those of the LNHL50 and LMK50 counterparts in the uptake phase. Similarly, in the release phase, the starting MBV values of the LCB10 are 14% and 42% higher than those of LNHL50 and LMK50, respectively. Moreover, the LCB10 shows approximately 1.2% higher steady-state MBV than LNHL50, and 18% higher than LMK50.

Figure 6 shows the percentage mass change relative to the initial mass during the test for the selected samples subjected to the humidity cycles. As presented, all specimens show the same mass change profile with a gradual increase or decrease, depending on the initial condition, until they reach a steady-state starting from the fourth cycle. Moreover, the LCB10 sample has the highest mass gain and release in the uptake and release portions of the cycle, respectively, regardless of the initial condition. The LNHL50 and LMK50 samples show comparable rates of mass gain and release.

Figure 7 presents the average steady-state MBV values of the hempcrete samples. The average steady-state MBV for all hempcrete samples of 2.78 gm/m^2^ RH%, with a standard deviation of 0.24 gm/m^2^ RH% (8.6%), falls in the range reported in the literature. For example, Collet et al. (2013) reported an MBV of 2.14 gm/m^2^ %RH, with a binder to hemp ratio of 2:1, and density of 430 kg/m^3^ [14], while Latif et al. (2015) reported an MBV of 3.47 gm/m^2^ %RH with a binder to hemp ratio of 1.2:1, and density of 290 kg/m^3^ [13]. The findings suggest that the initial moisture content condition does not significantly affect the final steady-state average MBV for each design mix (i.e., the difference is less than 1%).

## 4. Discussion

The overall findings suggest that hempcrete mixes with 1:1 binder to hemp hurd ratio and 300−400 kg/m^3^ density have mechanical and hygrothermal properties suitable for infill wall applications. For instance, the compressive strengths ranging from 0.09 to 0.57 MPa are comparable to the values reported by the previous studies that used higher binder to hurd ratios (i.e., 2:1 and 2.25:1) [18,26]. Similarly, thermal conductivity and specific heat capacity values in the range of 0.087−0.10 W/m K and 1250–1557 J/kg K correspond to the values reported by the studies that developed hempcrete samples with similar densities [5,10,26,46]. Furthermore, the average steady-state moisture buffer value (MBV) for all hempcrete samples of 2.78 gm/m^2^ RH indicates excellent moisture storage capabilities. The splitting tensile strength ranging from 0.0101 to 0.0348 MPa is the only measured parameter that is comparatively lower than the previous studies’ results. The likely reasons are the lower density range and the binder to this study’s hemp hurd ratio. In addition, hempcrete composites made from hydrated lime have significantly lower flexural strength than composites made from commercial composites [4,49].

The novel hempcrete mixes with hydrated lime and crushed brick (LCB) show comparable compressive and tensile strength to other design mixes, with the advantage of being composed of recycled material. For instance, the hempcrete samples with 10% crushed brick have compressive strengths higher than the average of all the design mixes. Additionally, the LCB samples have the lowest thermal conductivity considering their density and the highest average moisture buffer values for high and low initial moisture contents, which are approximately 18% higher than metakaolin formulas and comparable to design mixes with hydraulic lime.

The hempcrete’s density is a vital design parameter due to its significant effects on the material’s mechanical and thermal properties. For instance, on the one hand, hempcrete samples of approximately 340 kg/m^3^ and above have 45%–75% higher compressive and tensile strengths than those with lower density (i.e., 300−330 kg/m^3^). On the other hand, the low-density hempcrete samples (i.e., 290−310 kg/m^3^) have 8−13% lower thermal conductivity than those in the range of 320−336 kg/m^3^. Furthermore, as illustrated in Figure 8, an increase in thermal conductivity with a rise of compressive strength tends to be higher in the low than the upper range of compressive strength values. These findings indicate that the specific application should determine the density of the hempcrete biocomposites. The results also show that similar to previous studies [9,26], there is a strong positive linear correlation (R^2^ = 0.91) between the dry and wet samples’ conductivities and their densities. The relationship between compressive strength and density is weaker and increases over time from 0.36 at 28 days to 0.77 at 90 days.

The results also indicate that binder mix design and water content impact the prepared hempcrete samples’ thermal and mechanical properties. For example, an increase in the percent share of metakaolin and crushed brick reduces thermal conductivity and specific heat capacity. In contrast, increasing the hydraulic lime percentage increases the conductivity and specific heat capacity. In addition, an increase in the percent share of pozzolans above a certain amount can harm the mechanical properties of hempcrete, particularly when combined with reduced water content (i.e., 2.5:1 water to hemp ratio). This hindering effect is most significant when metakaolin is increased above 50%, followed by crushed brick above 10%.

## 5. Conclusions and Future Research

Despite excellent hemp-lime composites’ hygrothermal properties, their utilization in the construction industry remains low due to the high variability and lack of formulas with widely available ingredients. The use of innovative binders and a better understanding of the material’s performance are indispensable steps for addressing hempcrete’s underutilization. This is the first research study that applies recycled crushed brick as a pozzolan in hempcrete mixes to increase their availability and performance while reducing their environmental impact. Another contribution to the body of knowledge is the comprehensive mechanical and hygrothermal characterization of hemp-lime composites with different binder mix designs. Consequently, this study provides valuable information about insulating hemp-lime composites’ critical performance required for material characterization that is likely to interest audiences, in academia and industry, who are focused on sustainable low-carbon composite building materials. The main conclusions of this study that point to the need for future work and investigation are as follows:The thermal and mechanical properties of hempcrete depend on its density. Hence, the large-scale utilization of hemp-lime composites in the construction industry requires the use of manufactured blocks, wall panels, or spraying techniques to enable the construction of consistent code complying envelope systems. Therefore, future work should focus on the processing methods for optimizing and standardization of hempcrete design mixes for different envelope applications and installations. Furthermore, future work should increase the number of tested samples for each design mix to ensure consistency.The higher content of hemp hurd combined with an application-orientated level and direction of compaction can lead to the development of products suitable for infill wall utilizations. A follow-up study should investigate mechanical properties in the perpendicular and thermal behavior in the parallel direction of hempcrete formulas with a 1:1 binder to hemp hurd ratio and 300−400 kg/m^3^ density. Further research should also optimize compaction levels and orientations for infill wall installations of hemp-lime composites using blocks or wall panels.Although all the developed design mixes have mechanical and hygrothermal properties suitable for infill wall applications, in small amounts, recycled crushed brick is an excellent alternative to conventional pozzolans for creating more environmentally-friendly hempcrete composites. For example, the hempcrete samples with 10% crushed brick have the lowest thermal conductivity considering their density and the highest moisture buffer capacity. Furthermore, the new hydrated lime and crushed brick formulas have comparable mechanical properties to metakaolin and hydraulic lime formulas. Future research should combine and optimize the use of the investigated lime mortars and pozzolans in hemp-lime mixes. A follow-up study should also explore other locally available materials that can improve hemp-lime composites’ performance and affordability while reducing their environmental impact.The water content is an essential design parameter due to its significant impact on the hempcrete samples’ mechanical and thermal properties. This study indicates that a smaller amount of water (i.e., ≤2.5 kg) leads to some specimens’ inferior mechanical properties due to incomplete hydration caused by the high suction ability of the hemp hurd. A follow-up study should optimize water content for different infill wall application approaches such as blocks, wall panels, or spray-in. Further research should also develop a hemp pretreatment process and investigate water retainers’ application to balance the high suction of the hurd and improve mechanical behavior and durability of hemp-lime composites. Future research should also investigate hemp-lime composites’ durability and decay under different environmental conditions.


## Figures and Tables

**Figure 1 materials-13-05000-f001:**
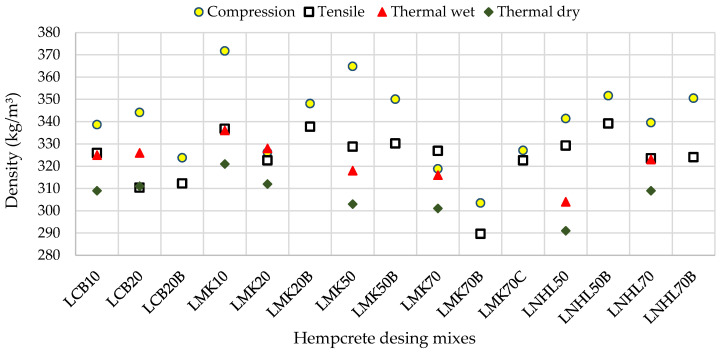
Average densities of all hempcrete samples.

**Figure 2 materials-13-05000-f002:**
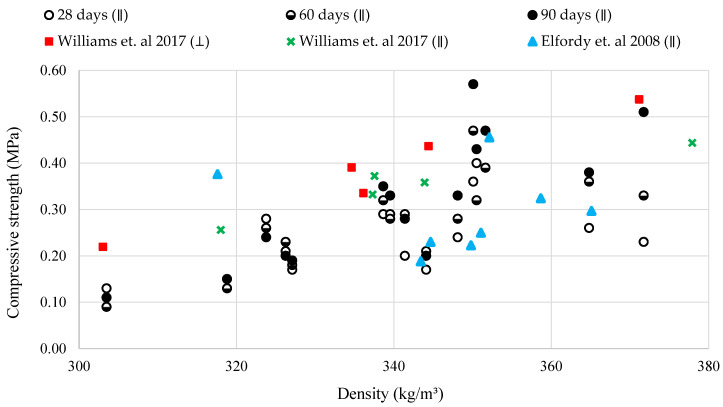
Comparison of density, compressive strength, and age.

**Figure 3 materials-13-05000-f003:**
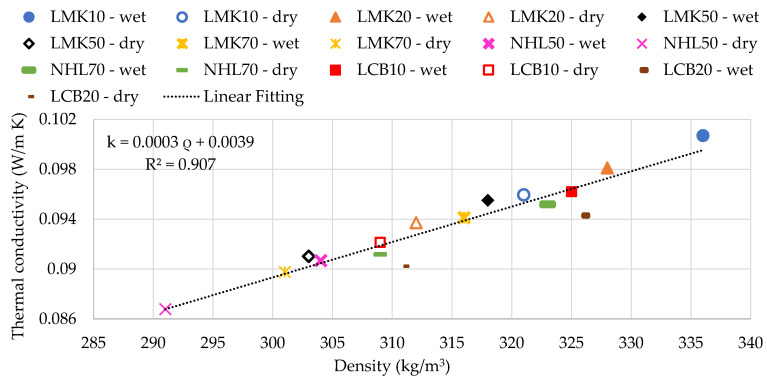
Thermal conductivity as a function of density.

**Figure 4 materials-13-05000-f004:**
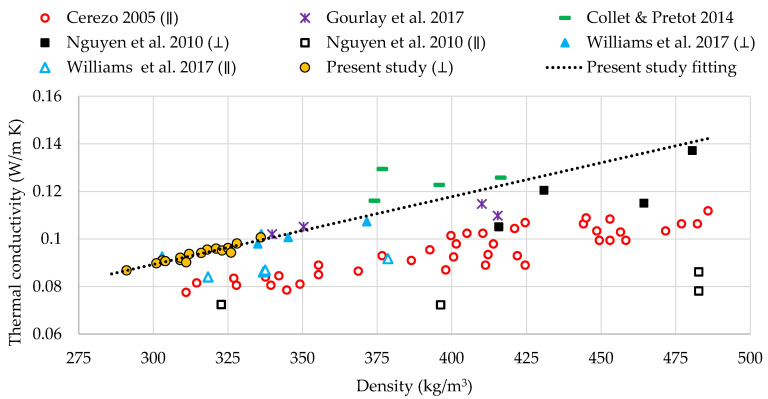
Thermal conductivity comparison with other studies.

**Figure 5 materials-13-05000-f005:**
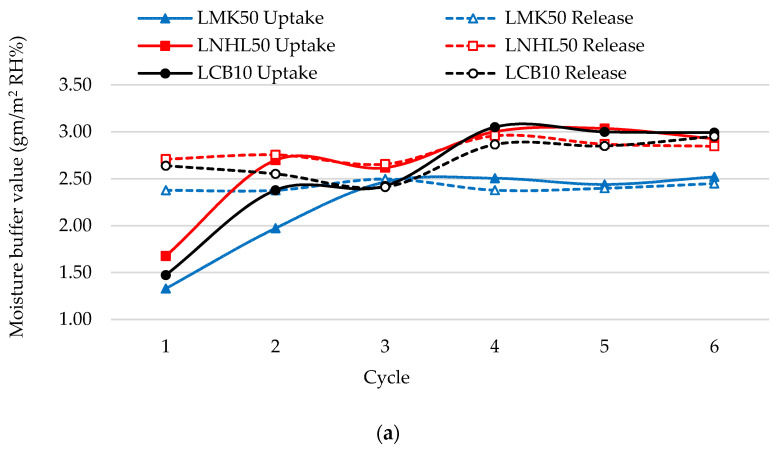
Moisture buffer value with cycles with different initial conditions. (**a**) High moisture content initial condition; (**b**) Low moisture content initial condition.

**Figure 6 materials-13-05000-f006:**
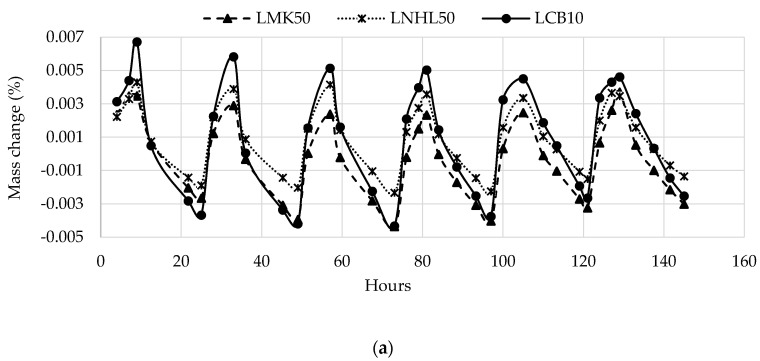
Typical steady cycle with different boundary conditions. (**a**) High moisture content initial condition; (**b**) Low moisture content initial condition.

**Figure 7 materials-13-05000-f007:**
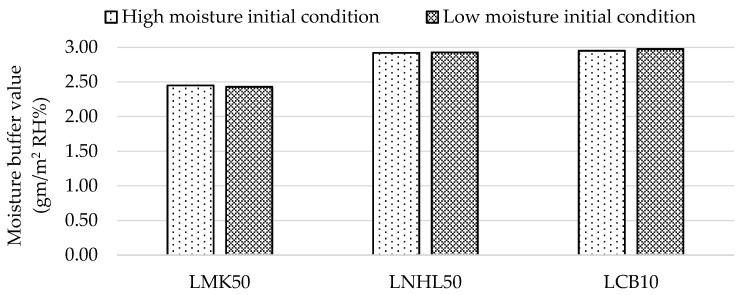
Average steady-state moisture buffer value (MBV).

**Figure 8 materials-13-05000-f008:**
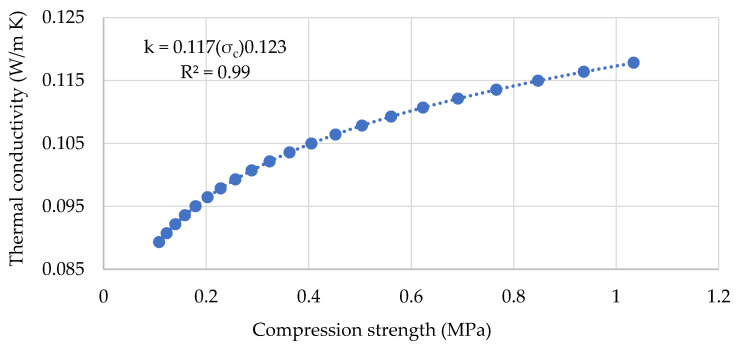
Thermal conductivity as a function of compressive strength.

**Table 1 materials-13-05000-t001:** Composition of hempcrete samples.

Design Mix	Name	Mix Ratio by Mass Hemp/Lime/Pozzolan/Water *						Hemp/Lime/Pozzolan/Water (by Mass)
		H	L	CB	MK	NHL	Water	
Hydrated lime-crushed brick	LCB10	1	0.9	0.1	-	-	2.5	22%/21%/2%/55%
	LCB20	1	0.8	0.2	-	-	2.5	22%/18%/4%/56%
	LCB20A	1	0.8	0.2	-	-	3	20%/16%//4%/60%
Hydrated lime-metakaolin	LMK10	1	0.9	-	0.1	-	2.6	22%//20%/2%/56%
	LMK20	1	0.8	-	0.2	-	3	20%/16%/4%/60%
	LMK20A	1	0.8	-	0.2	-	2.5	22%/18%/4%/56%
	LMK50	1	0.5	-	0.5	-	3	20%/10%/10%/60%
	LMK50A	1	0.5	-	0.5	-	2.5	22%/11%/11%/56%
	LMK70	1	0.3	-	0.7	-	2.5	22%/7%/15%/56%
	LMK70A	1	0.3	-	0.7	-	2.75	21%/6%/15%/58%
	LMK70B	1	0.3	-	0.7	-	3	20%/6%/14%/60%
Hydrated lime-natural hydraulic lime	LNHL50	1	0.5	-	-	0.5	2.5	22%/11%/11%/56%
	LNHL50A	1	0.5	-	-	0.5	3	20%/10%/10%/60%
	LNHL70	1	0.3	-	-	0.7	2.5	22%/7%/15%/56%
	LNHL70A	1	0.3	-	-	0.7	3	20%/6%/14%/60%

* H, hemp hurd; L, hydrated lime; MK, metakaolin; NHL, natural hydraulic lime; CB, crushed brick.

**Table 2 materials-13-05000-t002:** Descriptive statistics of samples.

Density (kg/m^3^)	Compression	Splitting Tensile	Thermal Wet	Thermal Dry
Minimum	304	290	304	291
Average *	340 ± 9	324 ± 6	322 ± 6	307 ± 6
Maximum	372	339	336	321
Standard deviation	17.2	12.1	8.9	8.3

* Confidence intervals (for 95% confidence level.).

**Table 3 materials-13-05000-t003:** Average compression strength, tensile strength, Young’s modulus, density, and water content.

Name	Water (kg)	Density (CT ^1^) (kg/m^3^)	Compressive Strength (MPa)	E (MPa)	Density (STT ^2^) (kg/m^3^)	Splitting Tensile Strength (MPa)
LCB10	2.5	339	0.32	10.42	326	0.0244
LCB20	2.5	344	0.19	6.38	310	0.0219
LCB20B	3	324	0.26	3.16	312	0.0223
LMK10	2.6	372	0.36	10.84	329	0.0219
LMK20	2.5	326	0.21	7.20	330	0.0270
LMK20B	3	348	0.28	9.00	323	0.0222
LMK50	2.5	365	0.33	8.05	338	0.0307
LMK50B	3	350	0.47	8.31	337	0.0342
LMK70	2.5	319	0.14	7.64	327	0.0160
LMK70B	2.75	304	0.11	4.27	290	0.0101
LMK70C	3	327	0.18	4.00	323	0.0183
LNHL50	2.5	341	0.26	9.01	329	0.0307
LNHL50B	3	352	0.42	5.21	339	0.0348
LNHL70	2.5	340	0.30	10.22	324	0.0260
LNHL70B	3	351	0.38	8.22	324	0.0229

^1^ Compression test; ^2^ splitting tensile test.

**Table 4 materials-13-05000-t004:** Thermal properties of the hempcrete samples.

Sample	State	Density (kg/m^3^)	Thermal Conductivity (W/m K)	Specific Heat Capacity (J/kg K)	Thermal Diffusivity (m^2^/s) × 10^−7^
LCB10	wet	325	0.0962	1504	1.968
	dry	309	0.0921	1350	2.209
LCB20	wet	326	0.0943	1398	2.069
	dry	311	0.0902	1250	2.321
LMK10	wet	336	0.1007	1536	1.951
	dry	321	0.0959	1397	2.140
LMK20	wet	328	0.0981	1530	1.955
	dry	312	0.0937	1385	2.169
LMK50	wet	318	0.0955	1523	1.972
	dry	303	0.0910	1379	2.178
LMK70	wet	316	0.0941	1498	1.988
	dry	301	0.0898	1352	2.206
LNHL50	wet	304	0.0907	1520	1.962
	dry	291	0.0868	1383	2.156
LNHL70	wet	323	0.0952	1557	1.893
	dry	309	0.0912	1421	2.076

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
