# Peer review of "Mechanical, Thermal, and Moisture Buffering Properties of Novel Insulating Hemp-Lime Composite Building Materials"

_materials, 2020, doi:10.3390/ma13215000_

Round 1

Reviewer 1 Report

The article deals with the important issue of using ecological and recovered building materials in new solutions of building composites.

The article is prepared at a good editorial and substantive level, and the obtained conclusions are supported by research results.

Notes for the authors of the article:

  1. In my opinion, the number of samples used to determine individual mechanical and thermal properties should be greater. In subsequent tests, I propose to reduce the number of prepared mixes, while increasing the number of samples to determine the parameter under study. In this case, the obtained results will be more precise.
  2. In Figure 2, on the horizontal axis, there is no description of the given feature, which is "density".
  3. In my opinion, the conclusion regarding the compressive strength of hempcrete (in lines 247 and 248) is true when using crushed brick and metakaolin, but not fully justified when using hydraulic lime (see results for LNHL50 and LNHL70).
  4. In the summary, it is worth trying to assess which or which of the prepared composite mixtures, in the Authors' opinion, obtained the most favorable distribution of the examined properties due to the possibility of application in real solutions for building partitions.

I recommend publishing the article after making some minor corrections.

Author Response

The article deals with the important issue of using ecological and recovered building materials in new solutions of building composites.

The article is prepared at a good editorial and substantive level, and the obtained conclusions are supported by research results.

  • Thank you very much for your time and valuable feedback. Our responses are provided on a point-by-point basis below. All changes within the manuscript are highlighted for better visibility.
  1. In my opinion, the number of samples used to determine individual mechanical and thermal properties should be greater. In subsequent tests, I propose to reduce the number of prepared mixes, while increasing the number of samples to determine the parameter under study. In this case, the obtained results will be more precise.
  • Future research will increase the number of samples and focus the study on specific design mixes that show favorable hygrothermal and mechanical properties. This limitation is now acknowledged; please see lines 485-486.
  1. In Figure 2, on the horizontal axis, there is no description of the given feature, which is "density".
  • Figure 2 has been modified. Please see the revised manuscript.
  1. In my opinion, the conclusion regarding the compressive strength of hempcrete (in lines 247 and 248) is true when using crushed brick and metakaolin, but not fully justified when using hydraulic lime (see results for LNHL50 and LNHL70).
  • This issue is now revised and clarified. Please see the revised manuscript.
  1. In the summary, it is worth trying to assess which or which of the prepared composite mixtures, in the Authors' opinion, obtained the most favorable distribution of the examined properties due to the possibility of application in real solutions for building partitions.
  • The paragraph in the conclusion section now addresses this; please see lines 490-500.

Reviewer 2 Report

The authors wrote an interesting paper and brought togehter an awfull lot of data on construction materials that have intersting properties, and might be usefull as sustainable construction materials.

I only have few remarks.

L312, 313 What is the error on the measurements? This holds for all measurements done in this work. I think the error is often larger than what would be expected from the numbers given. 4 significant digits for the heat capacity seems exagerated. Depending on the method and instrument the error will be about 5%, while 4 significant digits suggests an error of only 0.1%

And probably you have to reverse these numbers on line 313, to have the same order as before

L318 Can you explain why samples with more water have lower heat conduction rate? I would expect the opposite due to the mobility of water. On the other hand the heat capacity is higher.

The paper does not contain a separate conclusion, but a mixed discussion and conclusion. This part indeed contains a discussion with even an extra figure, and it contains several conclusions, so I insist to take out most of the conclusions and bring them togehter in one paragraph. Please be aware that people having limited time will only read the abstract and conclusions. Also when I ever read a paper and I would like to quickly check it again, I go to the conclusions and then maybe check a few figures. So the conclusion is of utmost importance.

Author Response

The authors wrote an interesting paper and brought together an awfull lot of data on construction materials that have interesting properties, and might be usefull as sustainable construction materials. I only have few remarks.

  • Thank you very much for your time and valuable feedback. Our responses are described on a point-by-point basis below. All changes within the manuscript are highlighted for better visibility.

L312, 313 What is the error on the measurements? This holds for all measurements done in this work. I think the error is often larger than what would be expected from the numbers given. 4 significant digits for the heat capacity seems exaggerated. Depending on the method and instrument the error will be about 5%, while 4 significant digits suggests an error of only 0.1%

  • The measurement error is now explained; please see 168-169 and 175-194. It is also worth mentioning that the heat flow meter presents the thermal conductivity results in the precision of 0.0001 (W/m K), the temperature in the precision of 0.01 oC, and heat flux in the accuracy of 0.01 W/m2.

And probably you have to reverse these numbers on line 313, to have the same order as before

  • The order is corrected; please see the revised version.

L318 Can you explain why samples with more water have lower heat conduction rate? I would expect the opposite due to the mobility of water. On the other hand, the heat capacity is higher.

  • Thermal tests included 'wet' samples at room temperature of 22±1 °C, with RH ~50% and 'dry' samples, oven-dried at 100±5 °C until reaching constant mass (i.e., changes in weight readings of less than 0.1%). This is now better explained; please see lines 166-168. In the wet state, the thermal conductivity and specific heat are higher than the dry state, as shown in Table 4. However, in the wet state, the total increase in heat storage (ρ x Cp) is higher than the increase in thermal conductivity (k). thus, the thermal diffusivityin a wet state is lower than that of the dry state. These results are now better explained; please see lines 329-332.

The paper does not contain a separate conclusion, but a mixed discussion and conclusion. This part indeed contains a discussion with even an extra figure, and it contains several conclusions, so I insist to take out most of the conclusions and bring them together in one paragraph. Please be aware that people having limited time will only read the abstract and conclusions. Also, when I ever read a paper and I would like to quickly check it again, I go to the conclusions and then maybe check a few figures. So, the conclusion is of utmost importance.

  • The conclusion section is now provided. Please see the revised version.

Reviewer 3 Report

The paper is very well-written and structured and presents in a clear and comprehensive way the parametric analysis was done on lab-scale samples of hempcrete in order to obtain the optimal mixing and casting process that leads to the sufficient mechanical and great thermal properties. The material proposed in this paper appears very promising and sustainable solution for renovation projects and the combination of hemp reinforcement with the addition of recycled brick material into the cement leads to a smart cementitious composite. The authors conducted an extended literature review and their findings are often related to the existing state-of-the-art. Abstract and Introduction sections are well written. Materials and Methods are also we reported. Great effort is paid at the Results section to correlate the different measuring variables and sophistically structure a multi-parametric analysis.

Minor syntax errors: Often a period is added after the citation of a figure, ie. ... presented in Figure 2. also show that....

Density in kg/m3 and the number 3 in superscript in all figures.

The paper can be accepted for publication in the current form.

Author Response

The paper is very well-written and structured and presents in a clear and comprehensive way the parametric analysis was done on lab-scale samples of hempcrete in order to obtain the optimal mixing and casting process that leads to the sufficient mechanical and great thermal properties. The material proposed in this paper appears very promising and sustainable solution for renovation projects and the combination of hemp reinforcement with the addition of recycled brick material into the cement leads to a smart cementitious composite. The authors conducted an extended literature review and their findings are often related to the existing state-of-the-art. Abstract and Introduction sections are well written. Materials and Methods are also we reported. Great effort is paid at the Results section to correlate the different measuring variables and sophistically structure a multi-parametric analysis.

  • Thank you very much for your time and valuable feedback. Our responses are provided on a point-by-point basis below. All changes within the manuscript are highlighted for better visibility.

Minor syntax errors: Often a period is added after the citation of a figure, ie. ... presented in Figure 2. also show that....

  • All syntax errors are revised and corrected.

Density in kg/m3 and the number 3 in superscript in all figures.

  • All figures' axes were revised and modified.

Reviewer 4 Report

This is a very interesting and cutting-edge material in the building sector, the Reviewer wants to congratulate the authors. However, there are some issues to resolve.

What regulations have been used to treat the workability of the samples? Has any specific experimental process been followed? It is important to highlight this property and its influence on the final resistance variations. 

Why were they only cured for seven days in a humid chamber and not for the 28 days prior to the trials?

How does hemp rot affect strength? Has the durability of these materials been considered?

Perhaps it is a good idea to combine the tests carried out in a single subsection, ahem: 2.4. Tests carried out, 2.4.1. Mechanical properties…

It may be interesting to build confidence intervals with the data from Table 2

In Figure 2, there does not appear to be much fit between variables. It may not make sense to show it or you need to make a better adjustment.

Some graphs of the results are not entirely clear and need to be detailed. Review axes and notation throughout the document.

What possible applications does this material have? Has it been verified with on-site or durability tests? It would be good to detail these possible limitations.

The conclusions must be reviewed.

Author Response

This is a very interesting and cutting-edge material in the building sector, the Reviewer wants to congratulate the authors. However, there are some issues to resolve.

  • Thank you very much for your time and valuable feedback. Our responses are provided on a point-by-point basis below. All changes within the manuscript are highlighted for better visibility.

What regulations have been used to treat the workability of the samples? Has any specific experimental process been followed? It is important to highlight this property and its influence on the final resistance variations. 

  • Currently, there is a lack of tests/regulations for measuring the workability of hemp-lime concretes, and experience is used to determine the appropriate water content. This is now acknowledged in the text; please see lines 126-127.

Why were they only cured for seven days in a humid chamber and not for the 28 days prior to the trials?

  • The samples were taken out from the molds after 7 days (i.e., after hardening), but the curation duration was from 21-28 for all samples until they reached a constant mass. Please see lines 142-144.

How does hemp rot affect strength? Has the durability of these materials been considered?

  • This study did not investigate the rotting and durability of the developed samples. Thus, we added a recommendation for future research. Please see lines 509-511.

Perhaps it is a good idea to combine the tests carried out in a single subsection, ahem: 2.4. Tests carried out, 2.4.1. Mechanical properties…

  • This is now addressed; please see the revised version.

It may be interesting to build confidence intervals with the data from Table 2

  • The confidence intervals are calculated for a 95% confidence level and added to Table 2. Please see the revised version.

In Figure 2, there does not appear to be much fit between variables. It may not make sense to show it or you need to make a better adjustment.

  • This is now addressed; please see the revised version.

Some graphs of the results are not entirely clear and need to be detailed. Review axes and notation throughout the document.

  • All figures and axes were revised and modified.

What possible applications does this material have? Has it been verified with on-site or durability tests? It would be good to detail these possible limitations.

  • The limitations, possible applications, and future research are added in the conclusion section. Please see the revised version.

The conclusions must be reviewed.

  • The conclusion section is added. Please see the revised version.